# Deep learning based automatic liver tumor segmentation in CT with shape-based post-processing

**Grzegorz Chlebus, Andrea Schenk, Jan Hendrik Moltz**
Fraunhofer MEVIS, Bremen, Germany
`{grzegorz.chlebus,andrea.schenk,jan.moltz}@mevis.fraunhofer.de`

**Bram van Ginneken**
Radboud UMC, Nijmegen, The Netherlands
Fraunhofer MEVIS, Bremen, Germany
`bram.vanginneken@radboudumc.nl`

**Horst Karl Hahn**
Fraunhofer MEVIS, Bremen, Germany
Jacobs University, Bremen, Germany
`horst.hahn@mevis.fraunhofer.de`

**Hans Meine**
University of Bremen, Germany
Fraunhofer MEVIS, Bremen, Germany
`meine@uni-bremen.de`

## Abstract

Accurate automatic liver tumor segmentation would have a big impact on liver therapy planning procedures and follow-up reporting, thanks to automation, standardization and incorporation of full volumetric information. In this work, we propose a fully automatic method for liver tumor segmentation in CT images based on a 2D convolutional deep neural network with a shape-based post-processing. We ran our experiments on the LiTS dataset and evaluated detection and segmentation performance. Our proposed method achieves segmentation quality for detected lesions comparable to a human expert and is able to detect 77% of potentially measurable tumor lesions according to the RECIST 1.1 guidelines. We submitted our results to the LiTS challenge achieving state-of-the-art performance.

## 1 Introduction

Liver cancer was the second most common cause of cancer-induced deaths in 2015 according to the World Health Organization. Hepatocellular carcinoma (HCC) is the most common type of primary liver cancer which is the sixth most prevalent cancer [1]. In addition, the liver is also a common site for secondary tumors. Liver therapy planning procedures would profit from an accurate and fast lesion segmentation, that allows for subsequent determination of volume- and texture-based information. Moreover, having a standardized and automatic segmentation method would facilitate a more reliable therapy response classification [2].

Liver tumors show a high variability in their shape, appearance and localization. They can by either hypodense (appearing darker than the surrounding healthy liver parenchyma) or hyperdense (appearing brighter), and can additionally have a rim due to the contrast agent accumulation, calcification or necrosis [3]. The individual appearance depends on lesion type, state, imaging (equipment, settings, contrast method and timing), and can vary substantially from patient to patient. This high variability makes liver lesion segmentation a challenging task in practice.

1st Conference on Medical Imaging with Deep Learning (MIDL 2018), Amsterdam, The Netherlands.

## 1.1 Related work

The problem of liver tumor segmentation has received a great interest in the medical image computing community. In 2008, the MICCAI 3D Liver Tumor Segmentation Challenge [4] was organized where both manual and automatic methods were accepted. Among the automatic ones, the best method applied an ensemble segmentation algorithm using AdaBoost [5]. Other submitted methods employed adaptive thresholding, region growing or level set methods [6–9]. In more recent years, methods using Grassmannian manifolds [10] and shape parametrization [11] were proposed.

Given the variability of liver lesions, a manual design of powerful features is not trivial. Deep neural networks (DNNs) gained rapidly growing attention in the computer vision community over the last years, because of their ability to learn features automatically from the data. Christ et al. [12] applied two cascaded U-net models [13] to the problem of liver and liver tumor segmentation. The approach employed one model solely for the liver segmentation and a separate one for the tumor segmentation within a liver bounding box. The final output was refined using a 3D conditional random field.

More recently, the Liver Tumor Segmentation (LiTS) challenge was organized [14]. All top-scoring automatic methods submitted to the two rounds organized in 2017 used DNNs. Han, the winner of the first round, used two U-Net like models with long and short skip connections, where the first model was used only for coarse liver segmentation allowing the second network to focus on the liver region. The second model was trained to segment both liver and tumors at once. The two models worked in 2.5D, i.e., they received five adjacent slices to segment the middle one, which provided the network with the 3D context information. The best method in the second LiTS round was developed by a group from Lenovo Research, China. Their approach employed two neural network ensembles for the liver and tumor segmentation, respectively. The ensembles consisted of U-Net models working in 2D and 2.5D trained with different hyperparameter settings. Other successful methods proposed to train jointly two networks for liver and tumor segmentation [15] and to exploit 3D information by training a 3D H-DenseUNet architecture using original image data as well as features coming from a 2D network [16].

## 1.2 Contribution

In the following, we describe a segmentation method for automatic liver tumor segmentation in CT based on a 2D deep neural network with shape-based post-processing. We propose to mask out voxels outside of the liver for the loss computation to concentrate the model on learning features distinguishing liver lesions from healthy liver parenchyma. Furthermore, we show that incorporation of 3D shape-based features allows for a significant reduction of false positives, which demonstrates the importance of 3D context information. Our proposed segmentation method achieved state-of-the-art performance in the second LiTS round.

## 2 Methods

### 2.1 Neural network

We decided not to use a 3D neural network architecture, because the slice thickness of CT scans used for training varies a lot. Resampling of the images to a homogeneous voxel size was not considered, because it could lead to removal of small lesions.

**Architecture**

We employed a modified U-net [13] fully convolutional network architecture (Fig. 1). Our model works on four resolution levels allowing for learning of local and global features. In the contracting (expanding) path convolutions (transposed convolutions) are used to decrease (increase) the spatial resolution and the feature map count is doubled (halved) with each transition. The network contains long skip connections passing feature maps from the contracting path to the expanding path allowing to recover fine details which are lost in the spatial downsampling. We also added short skip connections to have well-distributed parameter updates and to speed up the training [17]. Each convolutional layer uses 3x3 filter size and is followed by a batch normalization and a ReLU activation function. We used dropout (p=0.5) before each convolution in the upscaling path to prevent the network from overfitting.

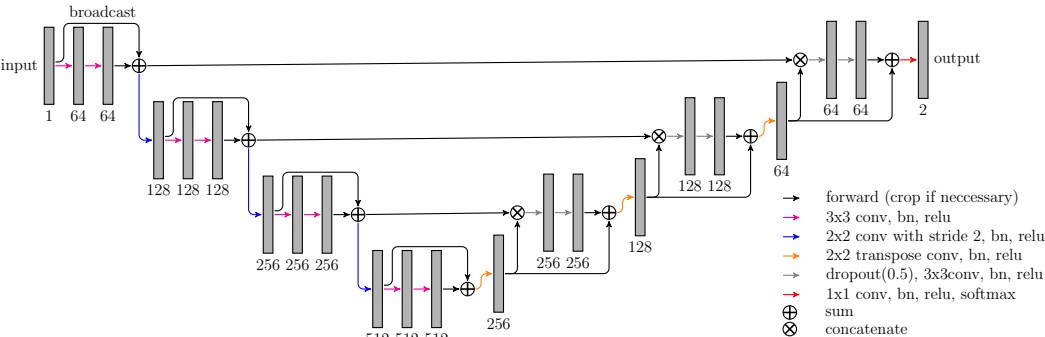

Figure 1: Overview of the neural network architecture. The numbers denote the feature map count.

### 2.1.1 Training

We trained the network using axial image patches in the original resolution and their corresponding labels. We used the soft dice coefficient as the loss function computed on the pixelwise softmax of the network final feature map [18]. The loss computation is restrained to a liver mask dilated by 10 mm in order to focus the model on the liver region. To deal with the high class imbalance, we ensured that each mini-batch contains patches where both classes (tumor and background) are present. We computed the parameter updates using the Adam optimizer with 5e-5 learning rate. The model was trained for 10 epochs (ca. 50k iterations, mini-batch size 6) with whole axial slices. We padded the input images with 44 pixels (reflect mode), because we used no zero-padding in the convolutions.

## 2.2 FP filtering

We observed that some of the neural network outputs corresponded to false positives, which could easily be identified due to their shape and location (e.g. liver/gallbladder boundary). Therefore, we extracted for each lesion 46 features using object-based image analysis [19] based on its shape, image intensity statistics and liver mask distance transformation. The latter was added in order to discriminate tumors at the border of the liver mask. The features were used to train a random forest classifier with 256 trees. Whether a tumor is true positive (TP) or false positive (FP) was determined using the evaluation code described in Sec. Evaluation.

## 2.3 Expert performance

In order to put the performance of our automatic method into a perspective, we asked a medical-technical radiology assistant (MTRA) with >10 years of segmentation experience to manually segment tumors in cases used for algorithm evaluation. This means that we have 2 reference annotation sets, which we refer to in the following as "MTRA" and "LiTS".

# 3 Experimental set-up

## 3.1 Data

In the following, we ran the experiments using the training dataset from the LiTS challenge containing 131 contrast-enhanced abdominal CT scans coming from 7 clinical institutions. The CT scans come with reference annotations of the liver and tumors done by trained radiologists. The in-plane resolution ranges from 0.5 to 1.0 mm and the slice thickness ranges from 0.7 to 5.0 mm. The dataset contains 908 lesions (63% with longest axial diameter $\geq$ 10 mm).

We divided the cases randomly into 3 non-overlapping groups for training, validation and testing containing 93, 6 and 30 cases, respectively. We removed 2 flawed cases due to missing reference tumor segmentation.

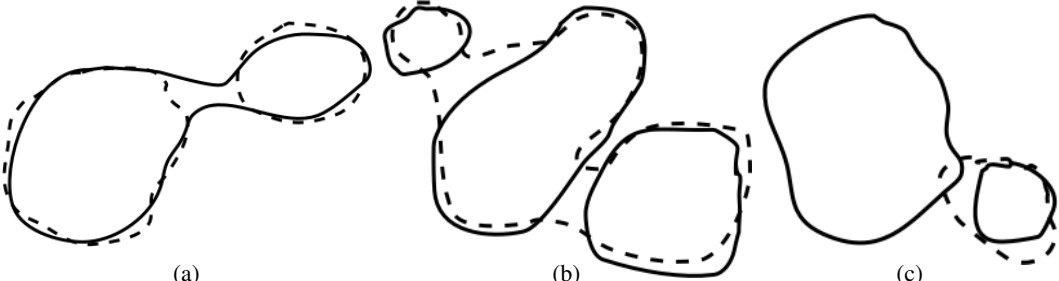

Figure 2: Non-trivial output(*dashed*)/reference(*solid*) correspondences. (a) Reference tumor corresponds to two output tumors (b) Three reference tumors correspond to one output tumor (c) Output tumor corresponds only to the smaller reference tumor.

## 3.2 Evaluation

We employ a similar evaluation methodology to the LiTS challenge measuring detection and segmentation performance. In contrast to LiTS, we use a method for determination of detected tumors, which does not require a lesion to be detected by only one contiguous object.

### 3.2.1 Detection

We evaluate the detection performance using metrics based on the Free-Response ROC analysis, which is suitable for experiments involving zero or more decisions per image [20]:

- Recall: Ratio of TP to real positives.
- Recall $\geq$ 10 mm: Ratio of detected tumors with longest axial diameter $\geq$ 10 mm to all such lesions in reference. The threshold value was derived from the RECIST 1.1 guidelines, where it is used to classify tumor lesions into measurable and non-measurable types [21].
- FPs/case: Count of FPs per case.

We define a hit, when the overlap (measured with the Dice index) between output and reference is above a threshold $\theta$ (0.2 in our case):

$$DICE\left(M^{\text{out}}[T^{\text{out}}], M^{\text{ref}}[T^{\text{ref}}]\right) > \theta$$

$M^{\text{out}}$ and $M^{\text{ref}}$ denote output and reference label images where each tumor has a unique label, $T^{\text{out}}$ and $T^{\text{ref}}$ are sets of output and reference tumor labels corresponding to each other. Notation $M[T]$ selects tumors with labels $T$ from $M$.

Determining output/reference tumor correspondence is not trivial, since situations as in Fig. 2 can occur. In Fig. 2a two output tumors $T^{\text{out}} = \{l_1^{\text{out}}, l_2^{\text{out}}\}$ correspond to one reference tumor $T^{\text{ref}} = \{l_1^{\text{ref}}\}$ and if their Dice index $> \theta$, then such situation should be counted as one TP. In Fig. 2b one output tumor $T^{\text{out}} = \{l_1^{\text{out}}\}$ corresponds to three reference tumors $T^{\text{ref}} = \{l_1^{\text{ref}}, l_2^{\text{ref}}, l_3^{\text{ref}}\}$ and if their overlap is above $\theta$, then such situation counts as three TP.

An algorithm for correspondence establishment of output/reference lesions should aim at maximizing the output/reference overlap. For example, consider Fig. 2c, where the output tumor should correspond only to the smaller reference tumor, since the overlap would decrease if both reference tumors would be considered. To account for $n : m$ correspondence situations where $n \neq m$, we count merge and split errors for each correspondence. Merge error is defined as $|T^{\text{ref}}| - 1$, split error as $\max(0, |T^{\text{out}}| - |T^{\text{ref}}|)$.

### 3.2.2 Segmentation

The segmentation quality was evaluated using the following measures:

- Dice/case: Computed by taking into account the whole output and reference tumor mask. When both masks are empty, a score of 1 is assigned.
- Dice/correspondence: Computed for each output/reference correspondence.

**Algorithm 1** Establishing correspondences between output and reference tumors.

```
 1: function CORRESPONDENCES(M^out, M^ref)
 2:     correspondences ← empty list
 3:     labels_out ← set of labels present in M^out
 4:     for all label in labels_out do
 5:         if label not present in M^out then
 6:             continue
 7:         candidates ← CANDIDATE_CORRESPONDENCES(M^out, M^ref, label)
 8:         s ← DICE(M^out[T^out], M^ref[T^ref]) for each (T^out, T^ref) in candidates
 9:         s_max ← max(s)
10:         if s_max > 0.2 then
11:             (T^out, T^ref) ← candidates[ARGMAX(s)]
12:             Append (T^out, T^ref) to correspondences
13:             Remove tumors with labels T^out from M^out
14:             Remove tumors with labels T^ref from M^ref
15:     return correspondences
16:
17: function CANDIDATE_CORRESPONDENCES(M^out, M^ref, l)
18:     T^ref ← ∅
19:     T^out ← {l}
20:     loop
21:         T^ref` ← set of tumor labels from M^ref overlapping with M^out[T^out]
22:         if T^ref` == T^ref then
23:             break
24:         T^ref = T^ref`
25:         T^out ← set of tumor labels from M^out overlapping with M^ref[T^ref]
26:     return (2^{T^ref} \ ∅) × (2^{T^out} \ ∅)
```

- Merge error: Sum of per correspondence merge errors.
- Split error: Sum of per correspondence split errors.

Algorithm 1 sketches the code we employed for establishing of correspondences between output and reference tumors.

## 4 Results and discussion

### 4.1 Expert performance

The MTRA needed 30-45 min. per case (the segmentation was done without time constraints). The comparison of MTRA with LiTS annotations and vice versa is shown in Tab. 1. The MTRA missed 11 of the LiTS lesions and found 78 additional ones, which accounts for 0.92 recall and 2.6 FP/case. The LiTS annotations identified correctly only 62% of tumors found by the MTRA. Smaller recall difference was observed for tumors $\geq 10$ mm, meaning that most of the lesions not included by the LiTS reference were small. The segmentation quality was 0.72 dice/correspondence and 0.7 dice/case. Fig. 3 shows example cases with major differences between MTRA and LiTS segmentations. There were two cases, where MTRA segmentation got 0 dice/case when compared with the LiTS reference: (i) a tumor was found in a case with no tumors, Fig. 3c, (ii) none of reference tumors were found.

Table 1: Comparison of segmentations done by human observers. The parentheses denote the dataset used as a reference for the computation of evaluation metrics.

| | Recall | Recall $\geq 10$ mm | FPs per case | Dice per case | Dice per correspondence | Merge error | Split error |
|---|---|---|---|---|---|---|---|
| MTRA (LiTS) | 0.92 | 0.94 | 2.6 | 0.70 | 0.72 | 11 | 5 |
| LiTS (MTRA) | 0.62 | 0.85 | 0.3 | 0.70 | 0.72 | 5 | 12 |

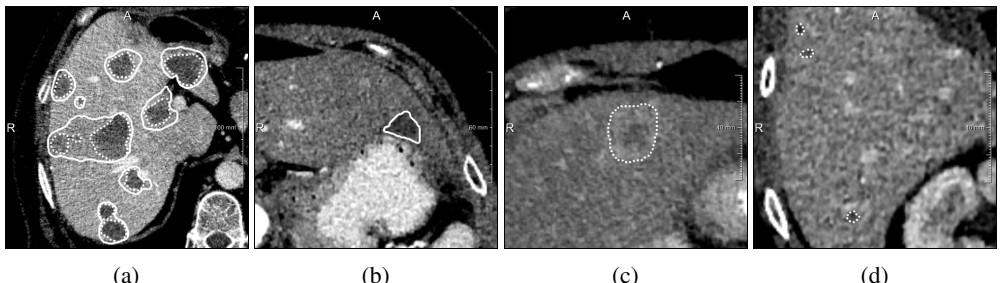

|       (a)       |       (b)       |       (c)       |       (d)       |

Figure 3: MTRA (*dashed*) vs. LiTS (*solid*) annotations. (a) Case with low dice/correspondence (b) Case where a LiTS reference tumor was missed (c) Case where MTRA found a lesion in a case with no tumors according to LiTS reference (d) Case where small additional tumors were found by the MTRA.

## 4.2   Neural network

The neural network was able to detect 47% and 72% of all tumors present in the MTRA and LiTS annotations, respectively. Tumors with longest diameter $\geq 10$ mm were detected more reliably than smaller ones (see Tab. 2); potentially measurable tumor lesions according to RECIST 1.1 had a recall of 75% and 86%, respectively. The false positive count was similar when comparing with MTRA and LiTS annotations (142 and 138, respectively). The segmentation quality was $\approx 0.5$ dice/case and $\approx 0.7$ dice/correspondence. 7 cases received 0 dice/case score (3 with no reference lesions and 4 where none of small reference lesions was found). Interestingly, the neural network, similar to the MTRA, found a lesion in the case with no tumors in the LiTS reference (Fig. 3c). Fig. 4 presents one example of a good segmentation produced by the neural network, as well as examples of different kinds of deviations from the reference.

Table 2: Mean metric values of investigated methods. Again, the parentheses denote the dataset used as a reference for the computation of evaluation metrics.

|                        | Recall | Recall $\geq 10$ mm | FP per case | Dice per case | Dice per correspondence | Merge error | Split error |
|------------------------|--------|---------------------|-------------|---------------|-------------------------|-------------|-------------|
| Neural network (MTRA)  | 0.47   | 0.75                | 4.7         | 0.53          | 0.72                    | 7           | 13          |
| Neural network (LiTS)  | 0.72   | 0.86                | 4.6         | 0.51          | 0.66                    | 12          | 14          |
| NN+FP filtering (LiTS) | 0.63   | 0.77                | 0.7         | 0.58          | 0.69                    | 11          | 10          |

## 4.3   FP filtering

We trained a random forest classifier on features computed for each tumor produced by the neural network from training and validation cases, where only LiTS annotations were available. Therefore, Tab. 2 reports results only for the LiTS reference. Among five most discriminative features four were shape-based (first eigenvalue, eccentricity, extent along z axis, voxel count). The remaining one described the std. deviation of the distance to the liver boundary. The classifier had 87% accuracy on test cases. 117 FPs were identified correctly, whereas 13 TPs (9 of which were $\geq 10$ mm) were wrongly rejected. The increase in dice/case was achieved by removing all FPs in two cases with no reference tumors.

## 4.4   Challenge submission and results

Before submission to the LiTS challenge, we trained the neural network further using all cases from the LiTS training dataset. Since our method requires liver masks, which were not given for the challenge test cases, we used our own liver segmentation method. We trained 3 orthogonal (axial, sagittal, coronal) U-Net models with 4 resolution levels for automatic liver segmentation on our in-house liver dataset from liver surgery planning containing 179 CTs [22]. We computed segmentations

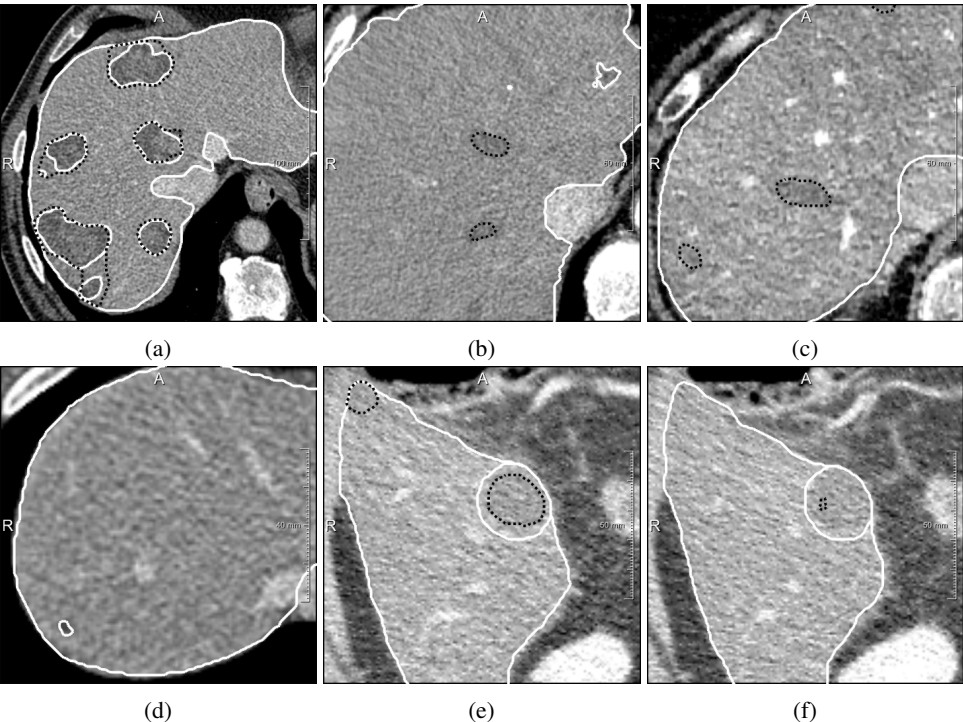

Figure 4: Neural network (*black*) compared with the LiTS (*white*) annotations. (a) Case with 0.85 dice/case (b,c) Cases with 19 and 16 FPs (d) Case where a small tumor was not detected (e,f) Case where tumor segmentation strongly differed on consecutive slices.

for the 70 challenge test cases ranking third at the MICCAI 2017 LiTS round (leaderboard user name `hans.meine`). Our submission scored 0.68 and 0.96 dice/case for tumor and liver segmentation, respectively. The tumor dice/case difference between our approach and the best submissions from MICCAI 2017 (`IeHealth`) and Open leaderboard (`xjqi` to date) is 0.02 and 0.04, respectively. Our method needs on average 67 s for one case: 43, 16 and 8 s for liver segmentation, tumor segmentation and FP filtering, respectively (Intel Core i7-4770K, 32GB RAM, GeForce GTX 1080).

## 5 Conclusions

In this work we described our method for automatic liver tumor segmentation in abdominal CT scans based on a 2D deep neural network, which ranked third in the second LiTS round at MICCAI 2017. We proposed to constrain the loss computation to a slightly dilated liver region and to employ a random forest classifier trained with hand-crafted features for false positive filtering. The fact that the most discriminative features in the FP filtering stage were shape-based indicates the importance of 3D information in distinguishing true from false positives.

Our method achieves segmentation quality for detected tumors comparable to a human expert and is able to detect 77% of potentially measurable tumor lesions in the LiTS reference according to the RECIST 1.1 guidelines. We observed that the neural network is capable of detecting bigger lesions (longest axial diameter $\geq 10$ mm) more reliably than smaller ones ($< 10$ mm). We presume, based on the performed comparison of LiTS annotations with those done by an experienced MTRA, that this can be attributed to a bigger inter-observer variability with respect to detection of smaller lesions. We think that the LiTS challenge data collection from multiple sites is a great initiative, that shows not only the variability in imaging, but also some variability in the annotations. This is probably due to the fact that liver tumor segmentation is not part of the daily routine, and that there are no universally agreed on clinical guidelines for this task.

We see the method described in this paper as promising, but it is clear that more work needs to be done to match the human detection performance. Moreover, an evaluation in a clinical setting will be

required to assess the clinical utility of automatic liver tumor segmentation methods. Future research directions include automation of reporting schemes for the liver.

### Acknowledgments

We gratefully thank Christiane Engel for annotating tumor lesions on 30 test cases used in this work.

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
