# OpenReview forum: "Deep learning based automatic liver tumor segmentation in CT with shape-based post-processing"
_MIDL.amsterdam/2018/Conference — Submitted to MIDL 2018_

### Review · AnonReviewer3 · 2018-05-06
**Decent evaluations comparing challenge ground truth, manual segmentation, and automated segmentation; but overall the paper is incremental with unclear contribution**

**Rating:** 1
**Confidence:** 3

**Review:**

In this paper the authors investigate segmentation of liver tumor from CT images, propose a weighted loss computation to handle the data imbalance issue, and a random forest post-processing to reduce false positives. In the evaluation, segmentations from LiTS challenge ground truth, a medical technical radiology assistant, and the proposed automated method are compared.

The contribution of this paper is rather unclear:
- The training of the proposed method requires additional liver masks in the training set (not clear how they are generated in this paper), however the tumor segmentation performance is inferior to the best performing entries which automatically segment both liver and tumor. Therefore the motivation of having the masks and using the proposed loss computation is not well-demonstrated.

- For the proposed random forest post-processing step, it is unclear what the features are used, if the features are extracted in 3D. Also, shape-based evaluations are necessary in order to show that the shape-based features are actually helpful (LiTS challenge adopts a few shape-based evaluation criteria, those could be considered in this paper).

-The algorithm for output/reference tumor correspondence establishment is interesting, it would be great to have further comments to link the evaluations to the clinical decision making/risk assessment in practice.


**Special Issue:**

No

---

> ### Comment · ~Grzegorz_Chlebus1 · 2018-06-13
> **Comment on features used to train the RF classifier**
>
>
> The features used for training of the random forest classifier are extracted in 3D. In the paper, we provided a list of five most discriminative features, which helped to reduce significantly the false positive rate (sec. 4.3). I agree, that a full list of extracted features would be helpful and should be included in the appendix.
>
> The reviewer wrote, that LiTS adopts few shape-based evaluation criteria. I don't see which of the LiTS evaluation criteria could assess any shape information. It would be really helpful, if the reviewer could kindly provide a list of criteria, that would help to assess the utility of the extracted features.

---

> ### Comment · ~Grzegorz_Chlebus1 · 2018-06-13
> **On liver masks, comparison with best entries and masked loss**
>
>
> The liver masks used for training of the tumor segmentation model are from the LiTS data set. I will make it more clear in text to avoid confusion with this regard.
>
> Our results reported in Tab. 2 were computed for a subset of the LiTS *training* data set, which was left out by us for evaluation purposes. A direct comparison of these results with the LiTS challenge submissions is not possible, because the results on the LiTS leaderboard were computed for the LiTS *test* data set. For that, we report in sec. 4.4 results of our method on the LiTS *test* data and compare them with best submissions from the MICCAI and Open leaderboard.
>
> Our evaluation of the impact of the masked loss was not finished before the conference deadline. We will report on the influence of the masked loss in the next paper revision.

---

### Review · AnonReviewer2 · 2018-05-07
**Nice manuscript, well described method, public data used.**

**Rating:** 4
**Confidence:** 2

**Review:**

The authors present a method for detecting and segmenting liver tumors from CT. They evaluate their method on the publicly available LiTS dataset. The method is well described. Contributions of the paper are made clear. Clear description of the evaluation metric for the lesion matching. Overall and interesting and high quality contribution.

Positive aspects of the manuscript:
- Public data used for training and evaluation
- Authors contribute additional reference standard and discuss differences between observers and automated method
- Method is well described

Negative aspects of the manuscript:
- None

Detailed comments:

2 Methods
Training: what was the size of the image patches? Later in the paragraph the use of "whole axial slices" is mentioned. This is confusing? Are the terms "patch" and "slice" used interchangeably?
Where did the liver masks used in training come from, where they set manually?

3 Experimental setup
Detection: definition for recall is confusingly worded, would rephrase to something like: "Ratio of TP detections to the number of positives in the reference standard"
FPs/case: Shouldn't this be the average nr of FPs per case?

Algorithm 1: nice job on making the complex matching algorithm clear!



**Special Issue:**

Yes

---

> ### Comment · ~Grzegorz_Chlebus1 · 2018-05-22
> **Response to detailed comments**
>
>
> Ad 2.
> The terms "slice" and "patch" were used interchangeably, which can be confusing (thank you for pointing this out). The training patch size was 512x512 and was chosen to cover whole axial slices.
> The liver masks used for loss computation of the tumor segmentation network came from the LiTS dataset. The liver masks used for training of liver segmentation networks required for the challenge submission were drawn manually by medical-technical radiology assistants.
>
> Ad 3.
> I agree, I will make sure to correct both definitions in the next revision of the paper.

---

### Review · AnonReviewer1 · 2018-05-09
**A solid paper but it lacks novelty**

**Rating:** 2
**Confidence:** 3

**Review:**

Overall:
The paper tackles a problem of automatic liver tumor segmentation in abdominal CT scans using a combination of a neural network (U-Net) and a random forest classifier. In general, the paper is very well-written and easy to follow. The obtained results are solid. The main drawback of the paper is its clear orientation on the MICCAI challenge. The paper combines well-known methods in an interesting manner, however, it does not introduce any novelty from the methodological point of view.

Strengths:
+ The method is well-thought and it is clear that the authors considered pros and cons of different architectures.
+ The considered problem is of great practical significance.
+ The proposed approach is well-suited to the problem.
+ The obtained results are impressive (the 3rd place in the MICCAI 2017 challenge).

Remarks:
* Major
- I have some doubts how the final structure of the model was chosen. After reading the following sentence: "We observed that some of the neural network outputs corresponded to false positives (...)", I wonder whether these observations were done basing on training or test data.
- Why did the authors pick a random forest classifier instead of other classifiers like SVM? What was the motivation for that?
- In general, the paper aims at a possibly best performance instead of developing a new method. Because of its pure engineering character, I decreased the overall assessment by 1 point.

* Minor
- Personally, I dislike that the authors add a random forest classifier on top of a trained neural network. It results in a two-stage learning algorithm where first a neural network is trained, and then a random forest is trained on extracted features from the neural network. However, this remark is really minor since the final results are in favor of the proposed approach. In general, it would be an interesting direction for future research to understand why a neural network alone cannot achieve similar or better results as the random forest classifier.

**Special Issue:**

No

---

> ### Comment · ~Hans_Meine1 · 2018-05-16
> **Rationale for CNN + Random Forest**
>
> The neural network works on slice-wise image data and classifies voxels. (For the given data, where the slice thickness varies a lot, and is often quite anisotropic, 2D works better here than 3D.) Classifying voxels means that the training data consists of voxels, which are orders of magnitudes more than volumes or slices, which makes it feasible to train robust networks. (Of course, nearby samples are not independent, but the statement still holds.)
>
> The random forest is used to classify *tumors*.  That's a whole different level, and the number of training samples is *much* lower.  At the same time, it allows to aggregate more information from the 3D tumor volume (image features, shape features, spatial context features).
>
> This is why it is not possible to achieve the same performance with a single classifier, and the choice of a Random Forest for the second stage is justified by the small amount of available training samples.

---

> ### Comment · ~Hans_Meine1 · 2018-06-12
> **Concerning "new method" vs. "pure engineering"**
>
> I would like to comment on the critique that "the paper aims at a possibly best performance instead of developing a new method".  Indeed it was our goal to solve this important, clinical task as good as possible, but we see this as a strength!
>
> An important part of scientific practice is solid evaluation, and we believe that there's currently an abundance of new methods already, which have not been exhaustively evaluated at all yet. Therefore, we don't see our contribution as "pure engineering", but would like to point out that our contributions included not only a state of the art segmentation approach, but also improved evaluation methods (via a definition of "true positive" adapted for lesions) and an experiment comparing our results with that of a human expert.
>
> We will check whether we can make our goals and contributions more clear in a future version of the paper.

---

### Decision · Program_Chairs · 2018-05-15
**Paper62 Acceptance Decision**

Reject